



# An improved model for air–sea exchange of elemental mercury in MITgcm-ECCOv4-Hg: the role of surfactants and waves

Ling Li[1], Peipei Wu[1], Peng Zhang[1], Shaojian Huang[1], Yanxu Zhang[1,2,]*

[1]School of Atmospheric Sciences, Nanjing University, Nanjing, Jiangsu 210023, China

[2]Frontiers Science Center for Critical Earth Material Cycling, Nanjing University, Nanjing, Jiangsu 210023, China

*Correspondence to: zhangyx@nju.edu.cn

**Abstract.** The air–sea exchange of elemental mercury ($Hg^0$) plays an important role in the global Hg cycle. Existing air–sea exchange models for $Hg^0$ have not considered the impact of sea surfactants and wave breaking on the exchange velocity, leading to insufficient constraints on the flux of $Hg^0$. In this study, we have improved the air–sea exchange model of $Hg^0$ in the three-dimensional ocean transport model MITgcm by incorporating sea surfactants and wave breaking processes through parameterization utilizing the total organic carbon concentration and significant wave height data. The inclusion of these factors results in an increase of over twofold in the transfer velocity of $Hg^0$ relative to the baseline model. Air–sea exchange flux is increased in mid- to high-latitude regions with high wind and wave breaking efficiency, while it is reduced by surfactant and concentration change at low latitudes with low wind speeds and nearshore areas with low wave heights. Compared with previous parameterizations, the updated model demonstrates a stronger dependence of $Hg^0$ air–sea exchange velocity on wind speed. Our results also provide a theoretical explanation for the large variances in estimated transfer velocity between different schemes.

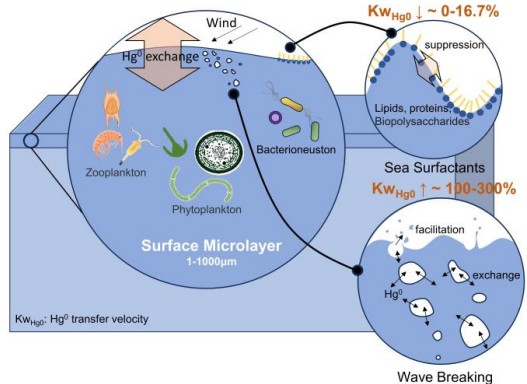

## 1 Introduction

Air–sea exchange of elemental mercury ($Hg^0$) contributes up to one-third of the total atmospheric



mercury (Hg) emissions. This process is crucial for the global Hg cycle, as it prolongs the residence time
of Hg in the biosphere (Amos et al., 2015) and reduces the reservoir of divalent mercury (Hg$^{II}$) in the
surface ocean (Lavoie et al., 2013). The air–sea exchange flux of Hg$^0$ is generally controlled by both
kinetic (gas transfer velocity, $k_{Hg}$) and thermodynamic (partial pressure related concentration gradients)
forcing (Wanninkhof, 1992; Wanninkhof et al., 2009; Kuss et al., 2011). However, the lack of direct
measurements of Hg$^0$ transfer velocity results in substantial uncertainty in estimating large-scale air–sea
Hg$^0$ exchange (Zhang et al., 2019). Considering that wind is the primary force driving turbulence in the
upper ocean, the transfer velocity is typically parameterized with wind speed through linear (Jähne et al.,
1979; Liss and Merlivat, 1986), quadratic (Wanninkhof et al., 1992; Nightingale et al., 2000), or cubic
relationships (McGillis et al., 2001; Edson et al., 2011). In addition, the gas transfer velocity is influenced
by other environmental factors such as surfactants and waves (Wurl et al., 2017). Therefore, relying
solely on wind speed may not be sufficient to quantify $k_{Hg}$.
Surfactants are ubiquitous in the sea surface microlayer (SML) and have associations with marine
biological production (Lin et al., 2002; Wurl et al, 2011). Surfactants are generally believed to affect air–
sea exchange in two ways: first, surfactants act as a physicochemical barrier that suppresses Hg$^0$ air–sea
exchange. Second, surfactants alter sea surface hydrodynamics, thus affecting turbulent energy transfer
(McKenna and McGillis, 2004; Engel et al., 2017), microscale fragmentation, and surface renewal
processes. Both experimental and modelling studies reveal that surfactants have a significant inhibitory
effect on the transfer velocity of various gases. Notably, a field experiment demonstrated that the
injection of artificial surfactant resulted in a suppression of transfer velocity (kw) by up to 55% (Salter
et al., 2011). Mesarchaki et al. (2015) observed that surfactants reduced the transfer velocity of $N_2O$ by
up to a factor of three in a large-scale wind-wave tank. Modelling research has shown that surfactants
could reduce global net $CO_2$ exchange by 15–50% (Asher, 1997; Tsai and Liu, 2003; Wurl et al., 2016).
Studies conducted by Kock et al. (2012) in the equatorial North Atlantic demonstrated an overestimation
of $N_2O$ using conventional kw methods, while the scheme considering the effect of surfactants (Tsai and
Liu, 2003) aligned well with the observations. Nevertheless, the impact of surfactants on the Hg$^0$ air–sea
exchange remains unknown.
Breaking waves produce bubbles that significantly facilitate the gas fluxes by increasing the air–water
interface and intensifying turbulence as the bubbles rise (Asher et al. 1996; Wanninkhof et al. 2009). This
is particularly pronounced for insoluble gases (Woolf and Thorpe, 1991; Kihm and Kortzinger, 2010;
Vagle et al., 2010). Woolf (1997) estimated that bubbles contribute 30% to the global $CO_2$ transfer
velocity, assuming a proportional relationship between bubble-mediated transfer velocity and whitecap
fraction. Historically, several models have been proposed to determine $CO_2$ exchange at the sea surface.
Zhang et al. (2006) found that the enhancement of gas transfer velocity for $O_2$ and $N_2$ due to bubbles can
be as high as 20%. According to Reichel and Deike (2020), approximately 40% of the net $CO_2$ flux
between the air and the ocean is attributed to bubbles. The importance of bubble effects depends on the
solubility of gases in seawater, and it is expected to be more significant for Hg$^0$ with lower solubility.
In this study, we have improved the MITgcm ocean model to gain a better understand of the mechanisms
that govern the air–sea exchange of Hg$^0$ at the atmosphere–ocean interface by including the effects of

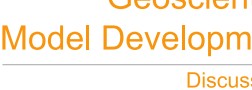
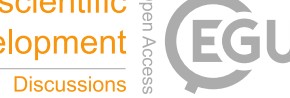

surfactants and wave breaking process. Sensitivity experiments are also conducted to analyze the effects
of individual factors on the $Hg^0$ transfer velocity. Additionally, we have examined the dependence of $Hg^0$
transfer velocity on wind speed.
**2 Methodology**
**2.1 MITgcm model**
The MITgcm (http://mitgcm.org/) is employed to simulate the air–sea exchange of $Hg^0$. We use a
configuration that has been fit to observations in a least-squares approach (ECCO v4; Forget et al., 2015).
This three-dimensional configuration features a horizontal resolution of 1°×1° and comprises 50 vertical
layers. Near the equator (0.5° latitude × 1° longitude) and the Arctic (approximately 40 km × 40 km), a
higher horizontal resolution is adopted to better simulate ocean currents. It calculates ocean physical
processes including vertical advection, diapycnal diffusion, and convective mixing based on ocean state
estimates from ECCO v4. The meteorological field of atmospheric variables (temperature, wind stress,
precipitation, humidity, and radiation) serves as the boundary layer of ocean are from the 6-hour ERA-
Interim reanalysis, spanning 1992 to 2017.
The model has the capacity to simulate the marine Hg cycles, which include the redox conversion
between $Hg^0$ and $Hg^{II}$, the methylation and demethylation of monomethylmercury ($CH_3Hg$) and
dimethylmercury [$(CH_3)_2Hg$], the air–sea exchange of $Hg^0$ and $(CH_3)_2Hg$, the partitioning between
dissolved and particulate mercury, the sinking of particulate-bound Hg, and the bioaccumulation of
$CH_3Hg$ in marine food webs (Zhang et al., 2014; 2020). Biogeochemical and ecological variables, such
as primary productivity (PP), particulate organic carbon (POC) and dissolved organic carbon (DOC) in
the ocean, are obtained from the Darwin marine ecosystem model (Dutkiewicz et al., 2012).
The baseline air–sea exchange of $Hg^0$ is calculated using the concentration gradient of $Hg^0$ across the
air–sea interface, the Henry's law constant (Andersson et al., 2008), the proportion of ice-free sea surface
areas, and the exchange velocity parameterized by wind speed proposed by Nightingale et al. (2000).
Additionally, the transfer velocity also depends on the temperature and salinity-corrected $Hg^0$ diffusion
rate in seawater (Wilke and Chang, 1955) and the temperature-corrected Schmidt number for $CO_2$
(Poissant et al., 2000). Based on the results of Loose et al. (2014), the exchange velocity for partially ice-
covered regions is doubled to accommodate the increased shear stress and convectively driven turbulence
induced by drifting sea ice. The $Hg^0$ air–sea flux ( $\text{Flux}_{Hg^0}$ ) is calculated as follows:
$$\text{Flux}_{Hg^0} = K_{Hg^0} \times (C_w - C_A / H) \qquad (1)$$

Where $C_w$ and $C_A$ represent the concentration of $Hg^0$ on the water and air side, respectively, H is the
Henry's law constant, which quantifies the ability of the dissolved phase to escape into the water, and
$K_{Hg^0}$ is the transfer velocity of $Hg^0$ on the ocean side calculated as follows:
$$K_{Hg^0} = (1 - iceo) \times pisvelo / \sqrt{Sc_{Hg^0} / Sc_{CO_2}} \qquad (2)$$





where
$$pisvelo = 0.222 \cdot u_{10}^2 + 0.333 \cdot u_{10} \tag{3}$$

Where $iceo$ is the sea ice coverage, $Sc_{Hg^0}$ and $Sc_{CO_2}$ are the Schmidt numbers for $Hg^0$ and $CO_2$,
respectively. $pisvelo$ is the piston velocity of $CO_2$ given by Nightingale et al. (2000).
The model is run from 1992 to 2011, allowing the response of Hg species to ocean physical and
biogeochemical changes to reach a steady state. The initial conditions are extracted from the previous
model output conducted by Zhang et al. (2020).
2.2 Parameterization of surfactants
Sea surface surfactant concentrations are related to PP, which is commonly represented by chlorophyll a
(Chl a) (Tsai and Liu, 2003). Nevertheless, recent studies have shown that Chl a cannot fully predict the
occurrence of surface surfactants when used as a substitute for PP (Wurl et al., 2011; Sabbaghzadeh et
al., 2017). Surface tension (Schmidt and Schneide, 2011), organic carbon concentration (Calleja et al.,
2009; Barthelmeß et al., 2021), and sea surface temperature (Pereira, 2018) are also used to predict the
occurrence of surface surfactants. However, most studies have not provided a clear quantitative
relationship. We adopt a relationship following Barthelmeß (2021) who found a linear relationship
between total organic carbon (TOC) and surface surfactant concentration in the Atlantic Ocean:
$$[SA] = 0.007[TOC] - 0.38 \tag{4}$$

where $[SA]$ represents the concentration of surface surfactants (mg TX-100 equiv. $L^{-1}$) and $[TOC]$
represents the concentration of TOC (μM) at the sea surface.
We model the influence of surfactants on piston velocity based on the empirical equation derived by
Pereira et al. (2018) from a shipboard gas exchange tank experiment in the Atlantic Ocean:
$$Suppression\ of\ kw[\%] = 32.44[SA] + 2.51 \tag{5}$$

where $Suppression\ of\ kw[\%]$ is the suppression of air–sea exchange velocity by surface surfactants.
Therefore, a parameterization (hereafter referred to as P18) was derived using the concentration of TOC
on the sea surface as an indicator of the suppression of air–sea exchange velocity by surface surfactants:
$$Suppression\ of\ kw[\%] = 0.227[TOC] - 9.817 \tag{6}$$

**2.2 Parameterization of wave breaking**
To take into account the effect of wave breaking on the air–sea exchange velocity, we separate the
contributions of wave breaking ($k_{bub}$) and non-breaking ($k_{int}$) following the approach of Woolf (2005).
The model agrees with measurements of $CO_2$ transfer at 20°C, but does not account for the dependence
of $k_{bub}$ on solubility. Therefore, this model is exclusively applicable to $CO_2$ and necessitates modifications
for $Hg^0$ compatibility (Jeffery et al., 2010). Here we take the influence of solubility into consideration.
For the non-breaking part, we utilize the squared wind speed parameterization (Nightingale et al., 2000)





previously adopted in the model:

$$k_{int} = pisvelo / \sqrt{Sc_{Hg^0} / Sc_{CO_2}} \qquad (7)$$

Regarding the wave breaking component, we attempt to use four different parameterization schemes. We
include significant wave height ($H_s$), which has been proved to be a more direct physical variable to
estimate air–sea exchange (Li et al., 2021). The Hs data we use are climatological monthly mean for the
2000–2020 obtained from ERA5 reanalysis data (Hersbach et al., 2020).
The first three parameterizations calculate the bubble-mediated transfer velocity as a function of whitecap
coverage:
Asher and Wanninkhof (1998), hereafter referred to as AW98:

$$k_{bub} = W_C \left( \frac{-37}{\alpha} + 6120\alpha^{-0.37} Sc_{Hg^0}^{-0.18} \right) \qquad (8)$$

Asher et al. (2002), hereafter referred to as A02:

$$k_{bub} = W_C \left( \frac{-37}{\alpha} + 10440\alpha^{-0.41} Sc_{Hg^0}^{-0.24} \right) \qquad (9)$$

Woolf (1997), hereafter referred to as W97:

$$k_{bub} = \frac{2450 W_C}{\alpha \left( 1 + \left( 14\alpha Sc_{Hg^0}^{-0.5} \right)^{-1/1.2} \right)^{1.2}} \qquad (10)$$

where $\alpha$ represents Ostwalt solubility (unitless), which is expressed according to Battino (1984) and
Andersson et al. (2008). $W_C$ represents the total whitecap coverage (unitless), encompassing both the
breaking crest generated by recent wave breaking (stage A whitecaps, $W_A$) and the sea surface foam in
the process of decay (stage B whitecaps, $W_B$). $W_A$ might be a better parameter for bubble-mediate transfer
velocity, owing to its more direct relationship with energy dissipation. Nevertheless, it exhibits weak
correlation with the Reynolds number and presents challenges in measurement. Therefore, we have opted
to employ the concept of total whitecap coverage for our calculations. It should also be pointed that, in
the case of AW98 and A02, we have focused exclusively on transfer via direct bubble exchange, which
provides a better simulate the transfer velocity (Blomquist et al., 2017). $W_C$ is a function of the wind sea
Reynolds number (RH, Woolf et al., 2005):

$$Wc = 4.02\times10^{-7} \times RH^{0.96} \qquad (11)$$

where

$$RH = \frac{u^* H_s}{\upsilon_\alpha} \qquad (12)$$

where u* is the friction velocity, and $\upsilon_\alpha$ is the air kinematic viscosity, with a value of $1.48\times10^{-5} \, m^2/s$ at
a temperature of 20℃.



The fourth parameterization utilizes a sea-state dependent gas transfer velocity parameterization
developed by Deike and Melville (2018), hereafter referred to as DM18. The DM18 parameterization is
based on direct numerical simulations of bubble dynamics beneath breaking waves (Deike et al., 2016),
as well as observations and modeling of wave and wave-breaking statistics (Deike et al., 2017). It has
been validated by field measurements of gas transfer velocity (Bell et al., 2017; Brumer et al., 2017):
$$k_{bub} = \frac{A_B}{\alpha}[u_*^{5/3}\sqrt{gH_s}^{4/3}] \tag{13}$$

where $A_B$ is dimensionless fitting coefficient ($A_B=1\pm0.2\times10^{-5}$ s$^2$ m$^{-2}$). The friction velocity (u*) is
represented by a piecewise linear function of the wind speed, as given by Edson (2013).
The expression for the air–sea exchange velocity, which takes into account the effects of surfactants and
wave breaking, is given by the following equation:
$$K_{wexch} = (1 - iceo) \times [k_{int} + k_{bub}] \times (1 - Suppression\ of\ kw[\%]) \tag{14}$$

Detailed parameterization and introduction of variables are listed in Table 1.
**Table 1.** Model parameterizations for wave breaking and surfactant

| Variables | Units | Description | Value or equation |
|---|---|---|---|
| Suppression of kw[%] | Unitless | Suppression of air–sea exchange velocity by surfactants | $Suppression\ of\ kw[\%] = 0.227[TOC] - 9.817$ |
| TOC | mol l$^{-1}$ | Sea surface total organic carbon concentration | TOC=DOC+POC |
| DOC | mol l$^{-1}$ | Sea surface dissolved organic carbon concentration | Darwin model |
| POC | mol l$^{-1}$ | Sea surface particle organic carbon concentration | Darwin model |
| k$_{bub}$ | m s$^{-1}$ | Bubble mediated gas transport rate | $k_{bub} = \frac{A_B}{\alpha}[u_*^{5/3}\sqrt{gH_s}^{4/3}]$ [a]  $\quad k_{bub} = \frac{2450W_C}{\alpha\left(1 + \left(14\alpha Sc_{Hg^0}^{-0.5}\right)^{-1/1.2}\right)^{1.2}} / 360000$ [b]  $\quad k_{bub} = W_C\left(\frac{-37}{\alpha} + 10440\alpha^{-0.41}Sc_{Hg^0}^{-0.24}\right) / 360000$ [c]  $\quad k_{bub} = W_C\left(\frac{-37}{\alpha} + 6120\alpha^{-0.37}Sc_{Hg^0}^{-0.18}\right) / 360000$ [d] |
| A$_B$ | s$^2$ m$^{-2}$ | Dimensional fitting coefficient [a] | $1\pm0.2\times10^{-5}$ |
| α | Unitless | Ostwalt solubility [e] | $\alpha = exp((-2404.3/t) + 6.92)$ |





| u[*] | m s[-1] | Friction velocity [f] | $u_* = \begin{cases} 0.03 \times u_{10}, & u_{10} < 4 \ m/s \\ 0.035 \times u_{10}, & 4 \ m/s < u_{10} < 8.5 \ m/s \\ 0.062 \times u_{10} - 0.28, & u_{10} > 8.5 \ m/s \end{cases}$ |
| Wc | Unitless | Total whitecap coverage factor [g] | $Wc = 4.02 \times 10^{-7} \times RH^{0.96} . RH = \dfrac{u^* H_s}{\upsilon_\alpha}$ |
| RH | Unitless | wind sea Reynolds number [g] | |
| $\upsilon_\alpha$ | m[2] s[-1] | Kinematic viscosity | $1.48 \times 10^{-5}$ |
| g | m s[-2] | Acceleration of gravity | 9.807 |
| $H_s$ | m | Significant wave height | ERA5 monthly data |

[a] Deike and Melville, 2018.
[b] Woolf et al., 1997.
[c] Asher and Wanninkhof, 1998.
[d] Asher et al., 2002.
[e] Battino, 1984; Andersson et al., 2008.
[f] Edson et al., 2013.
[g] Woolf et al., 2005

We conduct a total of eight simulations, including one baseline simulation, four simulations that
comprehensively consider the effects of wave breaking and surfactants: Case1 (P18 + DM18), Case2
(P18 + W97), Case3 (P18 + AW98), and Case4 (P18 + A02), and three sensitive experiments that solely
consider the effects of surfactants (Case5, P18) and wave breaking (Case6, DM18 and Case7, AW98).
**Table 2.** Experimental setting

| Parameterizations | Surfactants | Wave Breaking | | | |
|---|---|---|---|---|---|
| | P18[a] | DM18[b] | W97[c] | AW98[d] | A02[e] |
| Baseline | | | | | |
| Case1 | √ | √ | | | |
| Case2 | √ | | √ | | |
| Case3 | √ | | | √ | |
| Case4 | √ | | | | √ |
| Case5 | √ | | | | |
| Case6 | | √ | | | |
| Case7 | | | | √ | |

[a] Pereira et al., 2018
[b] Deike and Melville, 2018.
[c] Woolf et al., 1997.
[d] Asher and Wanninkhof, 1998.
[e] Asher et al., 2002.

**2.3 Observation Datasets**
We incorporate observational data from seven cruises that involved high-resolution synchronous
measurements of atmospheric and water $Hg^0$ concentrations in the Atlantic, Pacific and Southern Oceans.
These include data obtained by Kuss et al. (2011) during a transect from Punta Arenas, Chile, to
Bremerhaven, Germany, across the Atlantic in April–May 2009. Soerensen et al. (2013) reported data





from six cruises conducted between 2008 and 2010 in the Gulf of Maine, the New England Shelf, the
continental slope region and the Sargasso Sea. They also collected data along a latitudinal transect
(~20°N to ~15°S) in the central Pacific during the METZYME cruise in October 2011 (Soerensen et al.,
2014). Wang et al. (2017) obtained data during a cruise along the Antarctic coast from December 13,
2014 to February 1, 2015. Kalinchuk et al. (2020) reported data from a public cruise in the eastern Arctic
Ocean from September 7 to October 30, 2018. Mastromonaco et al. (2017) measured continuously in the
remote seas of western Antarctica, including Weddell Sea during winter and spring (2013) and
Bellingshausen, Amundsen and Ross seas during summer (2010/2011). All of these studies used similar
measurement methods, including Tekran trace mercury analyzers for atmospheric $Hg^0$ measurements and
automated continuous equilibrium systems for seawater $Hg^0$ measurements. The $Hg^0$ flux was calculated
based on a thin film gas exchange model (equation 1; Liss and Merlivat, 1986; Wanninkhof, 1992). The
transfer velocity was calculated using the Nightingale et al. (2000) or Wanninkhof (1992) parametrization
for instantaneous wind speeds, both characterized by a quadratic relationship with wind speed. The
reported data frequencies varied from 1 to 10 hours. Observational data on various forms of Hg
concentrations at the sea surface are summarized in Zhang et al. (2020).
**3 RESULTS AND DISCUSSION**
**3.1 Suppression of kw by surfactants**
Figure 1 presents the air–sea exchange velocity calculated by the baseline model and the suppression rate
of kw caused by the surface microlayer calculated from the annual average TOC concentrations. The
transfer velocity of baseline model is zonally distributed, with higher value (33.5 cm h$^{-1}$) at mid-to-high
latitudes, attributed to wind-induced turbulence enhancement. In this study, we term it as the transfer
velocity of non-breaking waves. Our parameterization of the suppression rate is directly related to the
distribution of DOC, which, in turn, is influenced by the biological activity (Hansell et al., 2009). The
model simulates a higher suppression rate in tropical and Arctic regions, reaching up to 16.7% (Fig. 1b),
but 5–10% in most regions. In tropical regions, organic matter resistant to degradation accumulates due
to vertical stratification. In Arctic regions, terrigenous organic matter is transported to the system via
high fluvial fluxes (Dittmar and Kattner, 2003). The lowest values are presented in the Southern Ocean,
where deep ocean waters are more readily mixed with the surface. Our estimated suppression effect of
surfactants generally aligns with Barthelmeß et al. (2021), who reported a suppression of kw of $CO_2$ by
11.5% (±SE 1.0) inside and 9.8% (±SE 2.2) outside the filament in the Atlantic Ocean. Similarly, Pereira
et al. (2018) found the kw suppressions reduced by 2 to 32% in the Atlantic in the presence of surfactants.
However, it is worth noting that other studies propose a greater impact. According to Pereira et al. (2016),
the exchange of $CO_2$ between the ocean and atmosphere decreased by 15 to 24% along the North East
coast of the UK. Furthermore, Yang et al. (2021) reported that the wind speed dependence of $CO_2$ transfer
velocity can vary by 30% in the Southern Ocean. Frew (1997) observed a fivefold reduction in gas
transfer velocity near the coast of New England due to increased surfactant abundance and DOC content
compared to the open ocean. Our lower estimate of the suppression effect might be reasonable, as their



samples were collected at different wind speed, which has significant role in surfactant suppression. The
highly variation in molecular composition across diverse environments also leads to a large variation in
surface activity (Barthelmeß et al., 2022). Therefore, the linear suppression relationship may change in
different environments. Additionally, some research conducted in the laboratory might not fully explain
processes in the natural environment (Krall and Jähne, 2014). To better explain the measured differences
in $Hg^0$ emissions between coastal and open ocean areas, we need to improve our understanding of how
surfactants and wind speed interact (e.g., marine aerosol emissions, surfactant abundance) to affect $Hg^0$
air–sea exchange velocity and subsequent net $Hg^0$ fluxes.

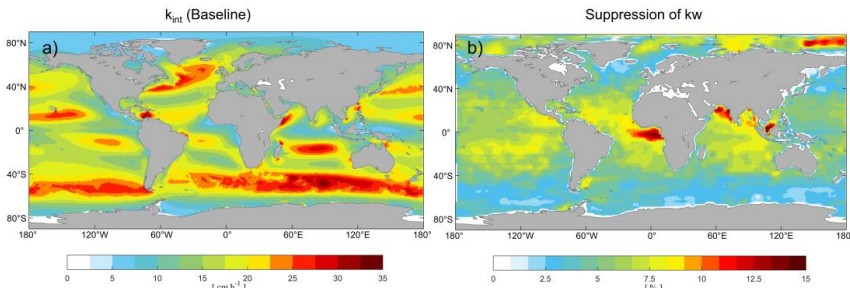


**Figure 1.** a) The annual mean non-breaking gas transfer velocity in unit of cm h⁻¹. b) The suppression of annual
mean $Hg^0$ gas transfer coefficient ($k_{Hg}$) by surfactants in unit of %.

### 3.2 Enhancement of kw by breaking wave

The bubble-mediated transfer velocities, calculated using four different bubble parameterizations, are
shown in Fig. 2. The spatial distribution of the velocities is quite similar for all the four scenarios with
relatively high values in regions with high wind speeds at mid- and high-latitudes, similar to the exchange
velocity of non-braking wave (Fig. 1a). However, the magnitude varies substantially among them. The
global mean bubble-mediated transfer velocities are 10.8, 9.9, 26.3 and 33.0 cm h⁻¹, respectively. Bubble-
mediated transfer velocities calculated with the DM18 parameterization (Fig. 2a) and the W97
parameterization (Fig. 2b) are comparable to those of non-breaking waves. Compared to the DM18
parameterization, the W97 parameterization shows less variation in exchange rates across latitudes, with
higher rates in low-latitude regions and lower rates in mid- and high-latitude regions. The reason may be
that DM18 have higher wind or wave height dependence of kw than that of W97 (Fig. S1). Conversely,
the AW98 (Fig. 2c) and A02 (Fig. 2d) parameterizations significantly enhance the air–sea exchange
velocity of $Hg^0$ (t test on means, p<0.001). In the Southern Ocean and the North Atlantic region, bubble-
mediated transfer rates can reach 105–120 cm h⁻¹, approximately 2–3 times higher than the transfer rates
of non-breaking waves. This can be explained by the employment of total whitecap coverage rather than
stage A whitecap ($W_A$), as $W_C$ is much higher than $W_A$ (Monahan and Woolf, 1989). Case 2–4 might
overestimate the bubble mediated transfer velocity. W97 was given for clean bubbles in quiescent water.
This parameterization ignores bubbles that are mixed to a considerable depth, leading to an
underestimation of the transfer velocity of poorly soluble gases (Woolf, 1997). AW98 and A02 have been
corrected for the dual-tracer method in laboratory simulations (Asher and Wanninkhof, 1998), but they
were not considered adequately for all cases, which is articulately important as gas transfer is highly



sensitive to void fraction (the ratio of air volume to total volume) and bubble plume (Woolf et al., 2007).
On the other hand, DM18 was developed by combining a mechanistic model for air entrainment and
bubble statistics with empirical relationships for wave statistics. It also has a good comparison with
measured and model data for different gases (Deike and Melville, 2018). In terms of physical
mechanisms, DM18 considers the process more comprehensively. Therefore, we suggest that DM18
might provide a better parameterization of wave breaking.
Our results demonstrate a higher contribution of wave breaking and bubbles to $Hg^0$ air–sea exchange
flux than $CO_2$. The bubble mediated transfer velocity in most regions is comparable with nonbreaking
transfer velocity, and it can reach up to 2–3 times as high as nonbreaking transfer velocity at high wind
speed region. But bubble transfer velocity of $CO_2$ accounts for a relatively small proportion in transfer
velocity according to previous studies (Woolf et al., 1997; Reichel and Deike, 2020). Woolf (1997)
estimated that bubbles contribute about 30% of the global $CO_2$ transfer velocity by assuming that the
transfer velocity mediated by bubbles is proportional to the coverage rate of whitecaps. Reichel and
Deike (2020) estimated that 40% of the $CO_2$ air–sea exchange fluxes in the Southern Ocean, North
Atlantic and Pacific are mediated by bubbles. This discrepancy could be attributed to gas solubility, as
the flux of less soluble gases is more enhanced by pressure effects (bubbles are compressed by hydrostatic
pressure) than more soluble gases (Bell et al., 2017; Reichel and Deike, 2020).

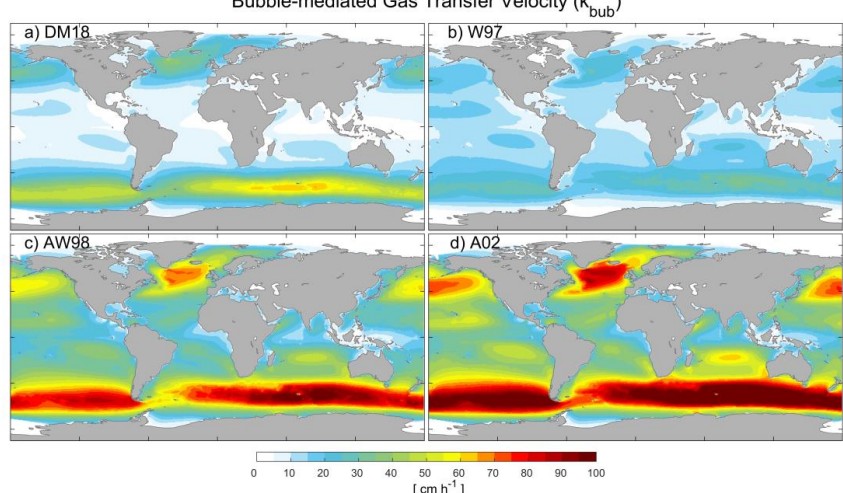


**Figure 2.** The annual mean bubble-mediated gas transfer velocity in unit of cm h$^{-1}$. The different bubble-mediated
parameterizations include a) DM18; b) W97; c) AW98 and d) A02.
**3.3 Wind speed dependence of $k_{Hg}$**
Most of the studies parameterize transfer velocity with 10 meter wind speed through linear
( $k_w = 2.8 \cdot u_{10} - 9.6$ , for 3.6< $u_{10}$ <13 m s$^{-1}$, Liss and Merlivat, 1986), quadratic
( $k_w = 0.222 \cdot u_{10}^2 + 0.333 \cdot u_{10}$ , Nightingale et al., 2000), or cubic relationships ( $k_w = 0.026 \cdot u_{10}^3 + 3.3$ ,
McGillis et al., 2001). Gaps among wind-based equations especially at developed wind-sea states cause





high uncertainty in different models. Recent research has shown that the transfer velocities of Hg⁰ have
a stronger dependence on wind speed by using eddy covariance flux measurements ( $k_w = 0.18 \cdot u_{10}{}^3$,
Osterwalder et al., 2021). Additional forcing factors, such as wave breaking and sea surface activators,
may result in different transport characteristics for different gases. In this section, sea surface temperature
(SST), TOC concentration and Hs of Case 1–4 are treated as random variables to fit the air–sea flux to
the 10-meter wind speed using power functions (Fig. 3):
$$\text{P18+DM18: } k_w = 0.181 \cdot u_{10}{}^{2.54}, r^2 = 0.893; \tag{15}$$
$$\text{P18+W97: } k_w = 0.362 \cdot u_{10}{}^{2.10}, r^2 = 0.963; \tag{16}$$
$$\text{P18+AW98: } k_w = 0.426 \cdot u_{10}{}^{2.33}, r^2 = 0.905; \tag{17}$$
$$\text{P18+A02: } k_w = 0.487 \cdot u_{10}{}^{2.34}, r^2 = 0.901 \tag{18}$$
Considering sea surface films and microscale wave breaking, the relationship between Hg⁰ exchange
velocity and wind speed appears to be between quadratic (Fig. 3a) and cubic (Fig. 3b and 3c), indicating
a stronger dependence than suggested by the typically used parameterizations (Nightingale et al., 2000;
McGillis et al., 2001). Compared with previous parameterizations (Fig. 3a and 3b), new
parameterizations (Fig. 3d–g, Fig. S2) show higher transfer velocity especially at high wind speeds, but
it is lower than that directly observed by Osterwalder et al. (2021; Fig. 3c) when wind speeds exceed 3–
5 m/s. The new parameterization suggests that bubble effects play an important role in boosting Hg⁰ air–
sea exchange and become more important at high wind speeds. Some previous parameterization schemes
may underestimate Hg⁰ emissions when wind speeds are high enough to induce wave breaking. In
comparison to gases with higher solubility such as $CO_2$, the air–sea exchange rate of Hg exhibits a
stronger dependence on wind speed, consistent with the findings of Osterwalder et al. (2021). Indeed,
microscale wave breaking enhances the transport velocity of poorly soluble gases, and bubble formation
is more effective at high wind speeds.

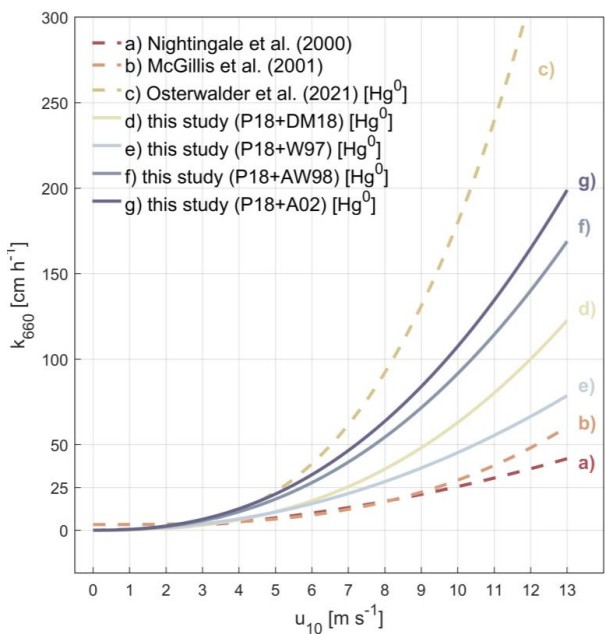

**Figure 3.** Wind speed dependence of transfer velocities used in gas exchange models to calculate air–sea fluxes. The k-values are normalized to Schmidt number of 660 (20 °C for $CO_2$ in seawater) and displayed against horizontal wind speed at 10 m [$u_{10}$]. For comparison, other wind speed relationships of the transfer velocity calculated by Nightingale et al. (2000), McGillis et al. (2001) and the cubic fit to measured transfer velocities of $Hg^0$ during two days of relaxed eddy accumulation $Hg^0$ emission measurements (Osterwalder et al., 2021) are included (dash line a–c). Solid curves d–g are the power fit to different model parametrization (Case 1–4). Case 1–4 have included the effect of wave breaking and surfactants. All four schemes employ the same surfactant parameterization DM18 and four different bubble parameterizations (DM18, W97, AW98 and A02).

### 3.4 $Hg^0$ exchange flux difference

The baseline model generally captures the spatial patterns of $Hg^0$ exchange flux (Fig. 4a), with lower flux in equator and polar regions and higher flux in mid-latitudes, which basically corresponds with the distribution of kw. Fig. 4b–d illustrates the simulated $Hg^0$ exchange fluxes by Case 5–7 compared with the baseline. The inclusion of the sea surfactant suppression effect alone results in a reduced flux in most areas, with the largest reduction in the North Atlantic, reaching -9% (Fig. 4b). However, the impact on a global level is minor, with only a 0.9% reduction in the global net Hg air–sea exchange flux compared with the baseline (3841 Mg $a^{-1}$), which equals to 3808 Mg $a^{-1}$. When only considering the effect of wave breaking (Fig. 4c and 4d), the exchange fluxes are estimated to be 4070 Mg $a^{-1}$ and 4189 Mg $a^{-1}$, respectively. Such values indicate an increase of 4.5% and 11.1% in global Hg exchange fluxes. The increased $Hg^0$ evasion may increase atmospheric Hg concentrations and thus Hg deposition and lifetime. Since only the oceanic part is considered in this model, i.e. $Hg^0$ deposition and atmospheric $Hg^0$ concentration as external forcing does not change with time, the increase in air–sea exchange fluxes significantly reduce the concentration of $Hg^0$ in the surface ocean (0–100 m; t test on means, p<0.001; Fig. S3), and thus alter other ocean Hg reservoirs (Fig. S4) and budgets (Fig. S5). This will result in an





augmentation of the magnitude of exchange flux changes, as effective bubble mediated transfer in the
regions of most developed wind-sea state significantly increase $Hg^0$ transfer velocity (t test on means,
p<0.001), while the impact of decreased concentration outweighs the slightly increased kw where the
waves are not well developed. As the result, the local variations of Case 5 and 6 range from -22.2% to
40.5% and -28.3% to 53.1%. We conclude that the model changes are primarily due to the inclusion of
bubble effect, whereas the inclusion of sea surface surfactants has a comparatively negligible impact on
the variations in air–sea exchange fluxes.
The global net fluxes based upon the combined effect of wave breaking and surfactants (Case 1–4) show
similar spatial patterns with baseline but present higher values (Fig. S6 and Fig. S7). The fluxes are 4056
Mg a$^{-1}$, 4016 Mg a$^{-1}$, 4155 Mg a$^{-1}$ and 4184 Mg a$^{-1}$, respectively, which are 5.6%, 4.6%, 8.2% and 8.9%
higher than the baseline (3841 Mg a$^{-1}$) because of the higher kw (Fig. S8). These values are also higher
than the estimates of 3360 Mg a$^{-1}$ by Zhang et al. (2023) and 3950 Mg a$^{-1}$ by Horowitz et al. (2017). The
local variations range from -21.8 to 39.5%, -16.2% to 28%, -28% to 51.3% and -30.7% to 56.2%,
respectively. However, all the modeled fluxes from Case 1 to Case 4 and baseline are within the large
uncertainty range of the observations, so we cannot determine which parameterization scheme provides
a more accurate estimate of air–sea exchange velocity simply by considering the current simulated results
in conjunction with the available flux observations. Indeed, the fluxes are highly sensitive to
concentration gradients and prevailing environmental conditions (wind speed, wave height and surfactant
concentration) with high-frequency temporal variability, modelling therefore could present rather general
zonal distribution (Fig. 4a and Fig. S6) than precise figures due to spatial and temporal resolution
limitations. For instance, during summer in Southern Ocean, the seawater can even be under-saturated,
leading to a net deposition of Hg from the atmosphere (Mastromonaco et al., 2017). This is not accurately
reflected in the annual mean flux modeled in our study. However, our study might explain why different
researches display great uncertainty in estimating $Hg^0$ exchange flux, as they ignored the effect of
surfactants and wave breaking. Therefore, further direct field measurements (especially micro-
meteorology techniques) are necessary to assess the transfer velocity of $Hg^0$, as well as the simultaneous
observation of surfactants and sea waves.

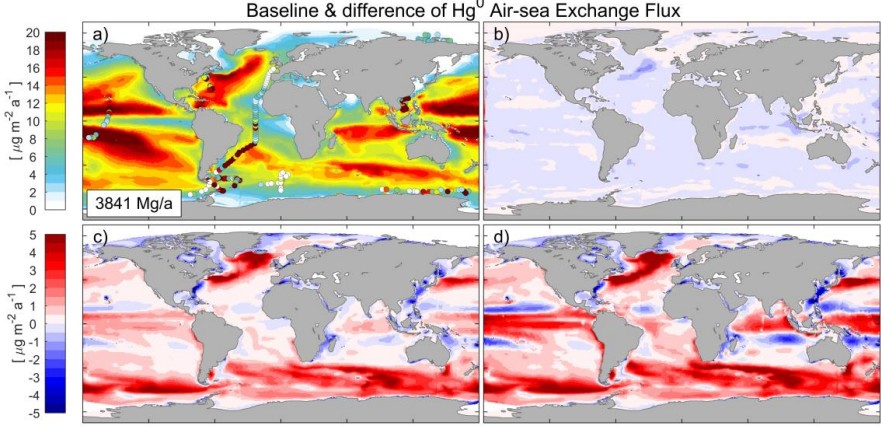


**Figure 4.** a) comparison between baseline model and observations (filled circles) for net $Hg^0$ air–sea exchange. Panels (b–d) are difference of annual mean net $Hg^0$ evasion flux with Baseline Model simulated by Case 5–7 which b) solely consider the effect of surfactant (Case5) with P18 parameterization and c) wave breaking with DM18 (Case 6) and AW98 (Case 7) parameterizations.

**3.5 Model Uncertainty**

The parameterization of the surfactant suppression is quite challenging, because changes in the chemical composition of surfactants may affect the relationship between TOC concentration and surfactant concentration (Barthelmess et al., 2022), as well as the inhibition relationship of the sea surface film (Mustaffa et al., 2019). Barthelmess et al. (2022) implied that refractory DOC from coastal land sources has a more persistent impact on air–sea exchange, while the inhibitory effect of semi-unstable organic matter (dissolved glucose and isoleucine) produced by phytoplankton is stronger but has a shorter impact time. The highly spatial-temporal variations in short-term and seasonal of surfactants and the chemical composition of the surfactant pool further increase uncertainty. On the other hand, wave breaking and bubble effect also show significant regional differences in the open sea and coastal waters (Callaghan et al., 2008; Woolf, 2005). The high-frequency temporal variability of the wind-wave processes and the limited resolution of wind-wave data used in this study may underestimate the variability caused by weather-scale $Hg^0$ transport. Currently, there is still a lack of quantitative research on the effects of different surfactant components and bubble effects on air–sea exchange. More detailed measurements of air–sea exchange velocity and related physical quantities are needed to better understand the importance of bubble-mediated and sea surface film-mediated Hg exchange. In addition, since only the ocean part is considered, the atmospheric $Hg^0$ concentration and deposition remain constant over time, which affect flux calculations to a substantially higher degree (Soerensen et al., 2013). To address this limitation, employing a coupled online model (Zhang et al., 2019), proves to be a valuable strategy for achieving a more accurate simulation of $Hg^0$ flux.

**4 Conclusion**

The estimation of $Hg^0$ air–sea exchange is of great uncertainty since only wind speed is the only parameter. Sea surfactants and breaking waves are thought to be two of the biggest drivers of uncertainty. In order to better assess the influence of surfactants and waves on $Hg^0$ air–sea exchange, we integrate sea surfactants and wave breaking processes into the air-sea exchange process of $Hg^0$ within the MIT General Circulation Model (MITgcm). Seven experiments (four combined experiments and three sensitive experiments) were conducted to explore the influence of sea surfactants and wave breaking on $Hg^0$ air–sea exchange flux.

We find that the $Hg^0$ transfer velocity can be suppressed by surfactants for 0–16.7%, while wave breaking contribute a much greater influence on it, as it is significantly increased 1–3 times because of low solubility of Hg. Therefore, we note that lack of consideration of these processes may lead to a vast underestimation of $Hg^0$ air–sea exchange flux. The new simulations that include sea surfactants and wave



anthropogenic emissions on the global mercury cycle: GLOBAL IMPACTS OF LEGACY MERCURY,
Global Biogeochem. Cycles, 27, 410–421, https://doi.org/10.1002/gbc.20040, 2013.
Amos, H. M., Sonke, J. E., Obrist, D., Robins, N., Hagan, N., Horowitz, H. M., Mason, R. P., Witt, M.,
Hedgecock, I. M., Corbitt, E. S., and Sunderland, E. M.: Observational and Modeling Constraints on
Global Anthropogenic Enrichment of Mercury, Environ. Sci. Technol., 49, 4036–4047,
https://doi.org/10.1021/es5058665, 2015.
Andersson, M. E., Gårdfeldt, K., Wängberg, I., and Strömberg, D.: Determination of Henry's law
constant for elemental mercury, Chemosphere, 73, 587–592,
https://doi.org/10.1016/j.chemosphere.2008.05.067, 2008.
Asher, W., Edson, J., Mcgillis, W., Wanninkhof, R., Ho, D. T., and Litchendor, T.: Fractional Area
Whitecap Coverage and Air-Sea Gas Transfer Velocities Measured During GasEx-98, in: Geophysical
Monograph Series, edited by: Donelan, M. A., Drennan, W. M., Saltzman, E. S., and Wanninkhof, R.,
American Geophysical Union, Washington, D. C., 199–203, https://doi.org/10.1029/GM127p0199, 2013.
Asher, W. E. and Wanninkhof, R.: The effect of bubble-mediated gas transfer on purposeful dual-gaseous
tracer experiments, J. Geophys. Res., 103, 10555–10560, https://doi.org/10.1029/98JC00245, 1998.
Asher, W. E., Karle, L. M., Higgins, B. J., Farley, P. J., Monahan, E. C., and Leifer, I. S.: The influence
of bubble plumes on air-seawater gas transfer velocities, J. Geophys. Res., 101, 12027–12041,
https://doi.org/10.1029/96JC00121, 1996.
Barthelmeß, T. and Engel, A.: How biogenic polymers control surfactant dynamics in the surface
microlayer: insights from a coastal Baltic Sea study, Biogeosciences, 19, 4965–4992,
https://doi.org/10.5194/bg-19-4965-2022, 2022.
Barthelmeß, T., Schütte, F., and Engel, A.: Variability of the Sea Surface Microlayer Across a Filament's
Edge and Potential Influences on Gas Exchange, Front. Mar. Sci., 8, 718384,
https://doi.org/10.3389/fmars.2021.718384, 2021.
Battino, R.: The Ostwald coefficient of gas solubility, Fluid Phase Equilibria, 15, 231–240,
https://doi.org/10.1016/0378-3812(84)87009-0, 1984.
Bell, T. G., Landwehr, S., Miller, S. D., De Bruyn, W. J., Callaghan, A. H., Scanlon, B., Ward, B., Yang,
M., and Saltzman, E. S.: Estimation of bubble-mediated air–sea gas exchange from concurrent DMS and
$CO_2$ transfer velocities at intermediate–high wind speeds, Atmos. Chem. Phys.,
17, 9019–9033, https://doi.org/10.5194/acp-17-9019-2017, 2017.
Blomquist, B. W., Brumer, S. E., Fairall, C. W., Huebert, B. J., Zappa, C. J., Brooks, I. M., Yang, M.,
Bariteau, L., Prytherch, J., Hare, J. E., Czerski, H., Matei, A., and Pascal, R. W.: Wind Speed and Sea
State Dependencies of Air-Sea Gas Transfer: Results From the High Wind Speed Gas Exchange Study
(HiWinGS), JGR Oceans, 122, 8034–8062, https://doi.org/10.1002/2017JC013181, 2017.
Brumer, S. E., Zappa, C. J., Blomquist, B. W., Fairall, C. W., Cifuentes-Lorenzen, A., Edson, J. B.,
Brooks, I. M., and Huebert, B. J.: Wave-Related Reynolds Number Parameterizations of CO 2 and DMS



Transfer Velocities, Geophys. Res. Lett., 44, 9865–9875, https://doi.org/10.1002/2017GL074979, 2017.
Callaghan, A., De Leeuw, G., Cohen, L., and O'Dowd, C. D.: Relationship of oceanic whitecap coverage
to wind speed and wind history, Geophys. Res. Lett., 35, L23609,
https://doi.org/10.1029/2008GL036165, 2008.
Calleja, M. L., Duarte, C. M., Prairie, Y. T., Agustı, S., and Herndl, G. J.: Evidence for surface organic
matter modulation of air-sea CO2 gas exchange, 2009.
Deike, L. and Melville, W. K.: Gas Transfer by Breaking Waves, Geophysical Research Letters, 45,
https://doi.org/10.1029/2018GL078758, 2018.
Dittmar, T. and Kattner, G.: The biogeochemistry of the river and shelf ecosystem of the Arctic Ocean: a
review, Marine Chemistry, 83, 103–120, https://doi.org/10.1016/S0304-4203(03)00105-1, 2003.
Dutkiewicz, S., Ward, B. A., Monteiro, F., and Follows, M. J.: Interconnection of nitrogen fixers and iron
in the Pacific Ocean: Theory and numerical simulations: MARINE NITROGEN FIXERS AND IRON,
Global Biogeochem. Cycles, 26, n/a-n/a, https://doi.org/10.1029/2011GB004039, 2012.
Edson, J. B., Fairall, C. W., Bariteau, L., Zappa, C. J., Cifuentes-Lorenzen, A., McGillis, W. R., Pezoa,
S., Hare, J. E., and Helmig, D.: Direct covariance measurement of $CO_2$ gas transfer velocity during the
2008 Southern Ocean Gas Exchange Experiment: Wind speed dependency, J. Geophys. Res., 116,
C00F10, https://doi.org/10.1029/2011JC007022, 2011.
Edson, J. B., Jampana, V., Weller, R. A., Bigorre, S. P., Plueddemann, A. J., Fairall, C. W., Miller, S. D.,
Mahrt, L., Vickers, D., and Hersbach, H.: On the Exchange of Momentum over the Open Ocean, Journal
of Physical Oceanography, 43, 1589–1610, https://doi.org/10.1175/JPO-D-12-0173.1, 2013.
Engel, A., Bange, H. W., Cunliffe, M., Burrows, S. M., Friedrichs, G., Galgani, L., Herrmann, H.,
Hertkorn, N., Johnson, M., Liss, P. S., Quinn, P. K., Schartau, M., Soloviev, A., Stolle, C., Upstill-
Goddard, R. C., Van Pinxteren, M., and Zäncker, B.: The Ocean's Vital Skin: Toward an Integrated
Understanding of the Sea Surface Microlayer, Front. Mar. Sci., 4, 165,
https://doi.org/10.3389/fmars.2017.00165, 2017.
Forget, G., Campin, J.-M., Heimbach, P., Hill, C. N., Ponte, R. M., and Wunsch, C.: ECCO version 4: an
integrated framework for non-linear inverse modeling and global ocean state estimation, Geosci. Model
Dev., 8, 3071–3104, https://doi.org/10.5194/gmd-8-3071-2015, 2015.
Frew, N. M.: The role of organic films in air–sea gas exchange, in: The Sea Surface and Global Change,
edited by: Liss, P. S. and Duce, R. A., Cambridge University Press, 121–172,
https://doi.org/10.1017/CBO9780511525025.006, 1997.
Hansell, D., Carlson, C., Repeta, D., and Schlitzer, R.: Dissolved Organic Matter in the Ocean: A
Controversy Stimulates New Insights, Oceanog., 22, 202–211,
https://doi.org/10.5670/oceanog.2009.109, 2009.
Hersbach, H., Bell, B., Berrisford, P., Hirahara, S., Horányi, A., Muñoz-Sabater, J., Nicolas, J., Peubey,
C., Radu, R., Schepers, D., Simmons, A., Soci, C., Abdalla, S., Abellan, X., Balsamo, G., Bechtold, P.,



Biavati, G., Bidlot, J., Bonavita, M., De Chiara, G., Dahlgren, P., Dee, D., Diamantakis, M., Dragani, R.,
Flemming, J., Forbes, R., Fuentes, M., Geer, A., Haimberger, L., Healy, S., Hogan, R. J., Hólm, E.,
Janisková, M., Keeley, S., Laloyaux, P., Lopez, P., Lupu, C., Radnoti, G., De Rosnay, P., Rozum, I.,
Vamborg, F., Villaume, S., and Thépaut, J.: The ERA5 global reanalysis, Quart J Royal Meteoro Soc,
146, 1999–2049, https://doi.org/10.1002/qj.3803, 2020.
Horowitz, H. M., Jacob, D. J., Zhang, Y., Dibble, T. S., Slemr, F., Amos, H. M., Schmidt, J. A., Corbitt,
E. S., Marais, E. A., and Sunderland, E. M.: A new mechanism for atmospheric mercury redox chemistry:
implications for the global mercury budget, Atmos. Chem. Phys., 17, 6353–6371,
https://doi.org/10.5194/acp-17-6353-2017, 2017.
Jahne, B., Münnich, K. O., and Siegenthaler, U.: Measurements of gas exchange and momentum transfer
in a circular wind-water tunnel, Tellus, 31, 321–329, https://doi.org/10.1111/j.2153-3490.1979.tb00911.x,
509  1979.

Jeffery, C. D., Robinson, I. S., and Woolf, D. K.: Tuning a physically-based model of the air–sea gas
transfer velocity, Ocean Modelling, 31, 28–35, https://doi.org/10.1016/j.ocemod.2009.09.001, 2010.
Kalinchuk, V. V., Lopatnikov, E. A., Astakhov, A. S., Ivanov, M. V., and Hu, L.: Distribution of
atmospheric gaseous elemental mercury (Hg(0)) from the Sea of Japan to the Arctic, and Hg(0) evasion
fluxes in the Eastern Arctic Seas: Results from a joint Russian-Chinese cruise in fall 2018, Science of
The Total Environment, 753, 142003, https://doi.org/10.1016/j.scitotenv.2020.142003, 2021.
Kihm, C. and Körtzinger, A.: Air‑sea gas transfer velocity for oxygen derived from float data, J. Geophys.
Res., 115, 2009JC006077, https://doi.org/10.1029/2009JC006077, 2010.
Kock, A., Schafstall, J., Dengler, M., Brandt, P., and Bange, H. W.: Sea-to-air and diapycnal nitrous oxide
fluxes in the eastern tropical North Atlantic Ocean, Biogeosciences, 9, 957–964,
https://doi.org/10.5194/bg-9-957-2012, 2012.
Krall, K. E. and Jähne, B.: First laboratory study of air–sea gas exchange at hurricane wind speeds, Ocean
Sci., 10, 257–265, https://doi.org/10.5194/os-10-257-2014, 2014.
Kuss, J., Zülicke, C., Pohl, C., and Schneider, B.: Atlantic mercury emission determined from continuous
analysis of the elemental mercury sea-air concentration difference within transects between 50°N and
50°S: ATLANTIC Hg SEA-AIR CONCENTRATION DIFFERENCE, Global Biogeochem. Cycles, 25,
n/a-n/a, https://doi.org/10.1029/2010GB003998, 2011.
Lavoie, R. A., Jardine, T. D., Chumchal, M. M., Kidd, K. A., and Campbell, L. M.: Biomagnification of
Mercury in Aquatic Food Webs: A Worldwide Meta-Analysis, Environ. Sci. Technol., 47, 13385–13394,
https://doi.org/10.1021/es403103t, 2013.
Li, S., Babanin, A. V., Qiao, F., Dai, D., Jiang, S., and Guan, C.: Laboratory experiments on CO2 gas
exchange with wave breaking, Journal of Physical Oceanography, https://doi.org/10.1175/JPO-D-20-
532  0272.1, 2021.

Lin, I.-I., Wen, L.-S., Liu, K.-K., Tsai, W.-T., and Liu, A. K.: Evidence and quantification of the



correlation between radar backscatter and ocean colour supported by simultaneously acquired in situ sea truth: CORRELATION BETWEEN RADAR BACKSCATTER AND OCEAN COLOUR, Geophys. Res. Lett., 29, 102-1-102–4, https://doi.org/10.1029/2001GL014039, 2002.

Liss, P. S. and Merlivat, L.: Air-Sea Gas Exchange Rates: Introduction and Synthesis, in: The Role of Air-Sea Exchange in Geochemical Cycling, edited by: Buat-Ménard, P., Springer Netherlands, Dordrecht, 113–127, https://doi.org/10.1007/978-94-009-4738-2_5, 1986.

Loose, B., McGillis, W. R., Perovich, D., Zappa, C. J., and Schlosser, P.: A parameter model of gas exchange for the seasonal sea ice zone, Ocean Sci., 10, 17–28, https://doi.org/10.5194/os-10-17-2014, 2014.

McGillis, W. R., Edson, J. B., Ware, J. D., Dacey, J. W. H., Hare, J. E., Fairall, C. W., and Wanninkhof, R.: Carbon dioxide flux techniques performed during GasEx-98, Marine Chemistry, 75, 267–280, https://doi.org/10.1016/S0304-4203(01)00042-1, 2001.

McKenna, S. P. and McGillis, W. R.: The role of free-surface turbulence and surfactants in air–water gas transfer, International Journal of Heat and Mass Transfer, 47, 539–553, https://doi.org/10.1016/j.ijheatmasstransfer.2003.06.001, 2004.

Mesarchaki, E., Kräuter, C., Krall, K. E., Bopp, M., Helleis, F., Williams, J., and Jähne, B.: Measuring air–sea gas-exchange velocities in a large-scale annular wind–wave tank, Ocean Sci., 11, 121–138, https://doi.org/10.5194/os-11-121-2015, 2015.

Monahan, E. C., and D. K. Woolf: Comments on "Variations of Whitecap Coverage with Wind stress and Water Temperature. J. Phys. Oceanogr., 19, 706–709, https://doi.org/10.1175/1520-0485(1989)019<0706:COOWCW>2.0.CO;2, 1989.

Mustaffa, N. I. H., Ribas-Ribas, M., Banko-Kubis, H. M., and Wurl, O.: Global reduction of in situ CO 2 transfer velocity by natural surfactants in the sea-surface microlayer, Proc. R. Soc. A., 476, 20190763, https://doi.org/10.1098/rspa.2019.0763, 2020.

Nerentorp Mastromonaco, M. G., Gårdfeldt, K., and Langer, S.: Mercury flux over West Antarctic Seas during winter, spring and summer, Marine Chemistry, 193, 44–54, https://doi.org/10.1016/j.marchem.2016.08.005, 2017.

Nightingale, P. D., Malin, G., Law, C. S., Watson, A. J., Liss, P. S., Liddicoat, M. I., Boutin, J., and Upstill-Goddard, R. C.: In situ evaluation of air-sea gas exchange parameterizations using novel conservative and volatile tracers, Global Biogeochem. Cycles, 14, 373–387, https://doi.org/10.1029/1999GB900091, 2000.

Osterwalder, S., Nerentorp, M., Zhu, W., Jiskra, M., Nilsson, E., Nilsson, M. B., Rutgersson, A., Soerensen, A. L., Sommar, J., Wallin, M. B., Wängberg, I., and Bishop, K.: Critical Observations of Gaseous Elemental Mercury Air-Sea Exchange, Global Biogeochemical Cycles, 35, https://doi.org/10.1029/2020GB006742, 2021.

Pereira, R., Schneider-Zapp, K., and Upstill-Goddard, R. C.: Surfactant control of gas transfer velocity



along an offshore coastaltransect: results from a laboratory gas exchange tank, Biogeosciences, 13, 3981–
3989, https://doi.org/10.5194/bg-13-3981-2016, 2016.
Pereira, R., Ashton, I., Sabbaghzadeh, B., Shutler, J. D., and Upstill-Goddard, R. C.: Reduced air–sea
CO2 exchange in the Atlantic Ocean due to biological surfactants, Nature Geosci, 11, 492–496,
https://doi.org/10.1038/s41561-018-0136-2, 2018.
Poissant, L., Amyot, M., Pilote, M., and Lean, D.: Mercury Water−Air Exchange over the Upper St.
Lawrence  River  and  Lake  Ontario,  Environ.  Sci.  Technol.,  34,  3069–3078,
https://doi.org/10.1021/es990719a, 2000.
Reichl, B. G. and Deike, L.: Contribution of Sea‐State Dependent Bubbles to Air‐Sea Carbon Dioxide
Fluxes, Geophys. Res. Lett., 47, https://doi.org/10.1029/2020GL087267, 2020.
Sabbaghzadeh, B., Upstill-Goddard, R. C., Beale, R., Pereira, R., and Nightingale, P. D.: The Atlantic
Ocean surface microlayer from 50°N to 50°S is ubiquitously enriched in surfactants at wind speeds up
to  13  m  s  −1:  Atlantic  Ocean  Surfactants,  Geophys.  Res.  Lett.,  44,  2852–2858,
https://doi.org/10.1002/2017GL072988, 2017.
Salter, M. E., Upstill‐Goddard, R. C., Nightingale, P. D., Archer, S. D., Blomquist, B., Ho, D. T., Huebert,
B., Schlosser, P., and Yang, M.: Impact of an artificial surfactant release on air‐sea gas fluxes during
Deep  Ocean  Gas  Exchange  Experiment  II,  J.  Geophys.  Res.,  116,  2011JC007023,
https://doi.org/10.1029/2011JC007023, 2011.
Schmidt, R. and Schneider, B.: The effect of surface films on the air–sea gas exchange in the Baltic Sea,
Marine Chemistry, 126, 56–62, https://doi.org/10.1016/j.marchem.2011.03.007, 2011.
Soerensen, A. L., Mason, R. P., Balcom, P. H., and Sunderland, E. M.: Drivers of Surface Ocean Mercury
Concentrations and Air–Sea Exchange in the West Atlantic Ocean, Environ. Sci. Technol., 47, 7757–
7765, https://doi.org/10.1021/es401354q, 2013.
Soerensen, A. L., Mason, R. P., Balcom, P. H., Jacob, D. J., Zhang, Y., Kuss, J., and Sunderland, E. M.:
Elemental Mercury Concentrations and Fluxes in the Tropical Atmosphere and Ocean, Environ. Sci.
Technol., 48, 11312–11319, https://doi.org/10.1021/es503109p, 2014.
Tsai, W.: An assessment of the effect of sea surface surfactant on global atmosphere-ocean CO 2 flux, J.
Geophys. Res., 108, 3127, https://doi.org/10.1029/2000JC000740, 2003.
Vagle, S., McNeil, C., and Steiner, N.: Upper ocean bubble measurements from the NE Pacific and
estimates of their role in air‐sea gas transfer of the weakly soluble gases nitrogen and oxygen, J. Geophys.
Res., 115, 2009JC005990, https://doi.org/10.1029/2009JC005990, 2010.
Wang, C., Wang, Z., Hui, F., and Zhang, X.: Speciated atmospheric mercury and sea–air exchange of
gaseous  mercury  in  the  South  China  Sea,  Atmos.  Chem.  Phys.,  19,  10111–10127,
https://doi.org/10.5194/acp-19-10111-2019, 2019.
Wang, J., Xie, Z., Wang, F., and Kang, H.: Gaseous elemental mercury in the marine boundary layer and



air-sea flux in the Southern Ocean in austral summer, Science of The Total Environment, 603–604, 510–
518, https://doi.org/10.1016/j.scitotenv.2017.06.120, 2017.
Wanninkhof, R.: Relationship between wind speed and gas exchange over the ocean, J. Geophys. Res.,
97, 7373, https://doi.org/10.1029/92JC00188, 1992.
Wanninkhof, R., Asher, W. E., Ho, D. T., Sweeney, C., and McGillis, W. R.: Advances in Quantifying
Air-Sea Gas Exchange and Environmental Forcing, Annu. Rev. Mar. Sci., 1, 213–244,
https://doi.org/10.1146/annurev.marine.010908.163742, 2009.
Wilke, C. R. and Chang, P.: Correlation of diffusion coefficients in dilute solutions, AIChE J., 1, 264–
270, https://doi.org/10.1002/aic.690010222, 1955.
Woolf, D. K.: Bubbles and their role in gas exchange, in: The Sea Surface and Global Change, edited by:
Liss,  P.  S.  and  Duce,  R.  A.,  Cambridge  University  Press,  173–206,
https://doi.org/10.1017/CBO9780511525025.007, 1997.
Woolf, D. K.: Parametrization of gas transfer velocities and sea-state-dependent wave breaking, Tellus
B: Chemical and Physical Meteorology, 57, 87, https://doi.org/10.3402/tellusb.v57i2.16783, 2005.
Woolf, D. K. and Thorpe, S. A.: Bubbles and the air-sea exchange of gases in near-saturation conditions,
J Mar Res, 49, 435–466, https://doi.org/10.1357/002224091784995765, 1991.
Woolf, D. K., Leifer, I. S., Nightingale, P. D., Rhee, T. S., Bowyer, P., Caulliez, G., De Leeuw, G., Larsen,
S. E., Liddicoat, M., Baker, J., and Andreae, M. O.: Modelling of bubble-mediated gas transfer:
Fundamental  principles  and  a  laboratory  test,  Journal  of  Marine  Systems,  66,  71–91,
https://doi.org/10.1016/j.jmarsys.2006.02.011, 2007.
Wurl, O., Wurl, E., Miller, L., Johnson, K., and Vagle, S.: Formation and global distribution of sea-
surface microlayers, Biogeosciences, 8, 121–135, https://doi.org/10.5194/bg-8-121-2011, 2011.
Wurl, O., Stolle, C., Van Thuoc, C., The Thu, P., and Mari, X.: Biofilm-like properties of the sea surface
and  predicted  effects  on  air–sea  CO2  exchange,  Progress  in  Oceanography,  144,  15–24,
https://doi.org/10.1016/j.pocean.2016.03.002, 2016.
Wurl, O., Ekau, W., Landing, W. M., and Zappa, C. J.: Sea surface microlayer in a changing ocean – A
perspective, Elementa: Science of the Anthropocene, 5, 31, https://doi.org/10.1525/elementa.228, 2017.
Yang, M., Smyth, T. J., Kitidis, V., Brown, I. J., Wohl, C., Yelland, M. J., and Bell, T. G.: Natural
variability in air–sea gas transfer efficiency of CO2, Sci Rep, 11, 13584, https://doi.org/10.1038/s41598-
021-92947-w, 2021.
Zhang, W., Perrie, W., and Vagle, S.: Impacts of winter storms on air-sea gas exchange, Geophys. Res.
Lett., 33, L14803, https://doi.org/10.1029/2005GL025257, 2006.
Zhang, Y., Jaeglé, L., and Thompson, L.: Natural biogeochemical cycle of mercury in a global three-
dimensional  ocean  tracer  model,  Global  Biogeochemical  Cycles,  28,  553–570,
https://doi.org/10.1002/2014GB004814, 2014.



Zhang, Y., Horowitz, H., Wang, J., Xie, Z., Kuss, J., and Soerensen, A. L.: A Coupled Global Atmosphere-
Ocean Model for Air-Sea Exchange of Mercury: Insights into Wet Deposition and Atmospheric Redox
Chemistry, Environ. Sci. Technol., 53, 5052–5061, https://doi.org/10.1021/acs.est.8b06205, 2019.
Zhang, Y., Soerensen, A. L., Schartup, A. T., and Sunderland, E. M.: A Global Model for Methylmercury
Formation and Uptake at the Base of Marine Food Webs, Global Biogeochem. Cycles, 34,
https://doi.org/10.1029/2019GB006348, 2020.
Zhang, Y., Zhang, P., Song, Z., Huang, S., Yuan, T., Wu, P., Shah, V., Liu, M., Chen, L., Wang, X., Zhou,
J., and Agnan, Y.: An updated global mercury budget from a coupled atmosphere-land-ocean model: 40%
more re-emissions buffer the effect of primary emission reductions, One Earth, 6, 316–325,
https://doi.org/10.1016/j.oneear.2023.02.004, 2023.