# Peer review of "An improved model for air-sea exchange of elemental mercury in MITgcm-ECCOv4-Hg: the role of surfactants and waves"

_Geoscientific Model Development, 2024_

## Author Response (AR1)

**Dear editor and dear reviewers,**

**Thank you for your letter and the reviews' comments concerning our manuscript entitled "An improved model for air–sea exchange of elemental mercury in MITgcm-ECCO v4-Hg: the role of surfactants and waves" (gmd-2024-81). These thoughtful comments are valuable and helpful for improving our paper. We have studied the comments carefully and provided our responses to the referee's comments below (in blue).**

**Response to Referee #1:**

The air-sea exchange velocity of elemental mercury, one of the major factors of uncertainty in the global mercury cycle, is still most commonly parametrized by using the 10-m wind speed only. The manuscript by Li et al. uses a new parametrization for the air-sea exchange velocity of mercury, also taking into consideration the exchange-suppressing effects of surfactants using total organic carbon concentrations, and the exchange-enhancing effects of wave breaking and bubble injection using the significant wave height. Using the MITgcm ocean model, one parametrization for the exchange suppression by surfactants and four different parametrizations for the exchange enhancement by wave breaking are tested and compared. The relative importance of each effect is constrained through several sensitivity runs. Overall, higher net oceanic elemental mercury evasion is obtained globally by the inclusion of these new parametrizations, when compared to the baseline run. The study also proposes that the non-consideration of surfactants and wave-breaking effects might be the reason why different observational studies obtained very different results for the relationship between exchange velocity and wind speed.

The article is generally well and concisely written, although the conclusion section might need a bit of retouching. All necessary details are given and the provided summary tables are appreciated. The argument is logically developed, and the overall line of thought is easy to follow. Relevant previous work is adequately referenced. This study fits well into the journal's scope, and the topic is interesting and of significant relevance for the mercury community. I also salute the authors for communicating the needs of the modelling community to experimentalists. Nevertheless, before suggesting publication I would like some of my doubts to be dispersed and some minor points to be addressed.

We extend our heartfelt appreciation for your professional review of our article. Your constructive and insightful feedback and suggestions are truly valued. To effectively address the issues you raised, we have undertaken extensive revisions to our initial draft.

**1. comment:**
Section 2.2 & general approach:
Woolf (2005) indeed separated the air-sea exchange velocity into contributions of wave breaking and non-wave breaking. However, for the non-wave breaking part, they state to have used a relationship derived from theoretical considerations and observations in wind-wave tanks. Thus, their representation of the non-wave breaking part, while most likely being influenced by microscale wave breaking as they explicitly state, was probably not influenced by larger scale wave breaking and significant bubble injection. In consequence, the addition of the non-wave-breaking term and another wave-breaking term likely led to no significant "double counting" of effects.

In contrast, in the present work, the authors use for the non-wave breaking part the parametrization of Nightingale (2000), which is based on deliberate tracer experiments in the southern North Sea, i.e. experimental data in the real ocean, spanning a wide range of different wind speeds (and thus likely including some wave-breaking). As such, it is likely that the parametrization of Nightingale already implicitly includes some effect of wave-breaking and bubble injection on the air-sea exchange velocity, even though this is not explicitly quantified.

I fear that by adding to the non-wave breaking part of the air-sea exchange velocity (using the parametrization of Nightingale (2000)) another term that explicitly considers the wave-breaking part, one risks counting the effect of wave-breaking and bubble injection twice, so to speak. This would lead to an overestimation of the total air-sea exchange velocity, which could explain why an overall global increase in the air-sea exchange velocity and the magnitude of oceanic Hg evasion was found in the present work.

I would like the authors to comment on this potential issue.

Your question is greatly appreciated. We have compared the parameterization of Nightingale with other non-breaking parameterizations. Deike and Melville (2018) estimated the nonbreaking transfer velocity using the COARE 3.1 parameterization (Fairall et al., 2011), close to the one proposed independently by Jahne et al. (1987) used in Woolf (2005).

$$k_{non-breaking} = 1.55 \times 10^4 \, u^* \quad \text{(Deike and Melville, 2018)}$$

$$k_{non-breaking} = 1.57 \times 10^4 \, u^* \quad \text{(Woolf et al., 2005)}$$

Asher et al., (2002) separated transfer velocity into three components, including bubble-mediated transfer ($W_C k_B$), turbulence transfer velocity generated by the breaking wave ($W_C(k_T - k_M)$) and the transfer velocity due to turbulence generated by all other processes ($k_M$). Here we regarded $k_M$ as nonbreaking transfer velocity.

$$kw = [k_M + W_C(k_T - k_M)] + W_C k_B \quad \text{(Asher et al., 2002)}$$

$$k_M = 47u \cdot Sc^{-0.5}$$

(1) Nightingale et al. (2000) assumed the transfer velocity among different gases only depend on diffusivity to the power -0.5 without taking solubility effect into consideration, which may only apply to non-breaking waves. When breaking waves make a significant contribution to gas exchange, like $Hg^0$, this relationship may not be applicable.

(2) The advantage of our theme is that it can distinguish the non-wave from the wave part, since there are very different functional forms between the two and the wind speed (e.g. linear or quadratic for non-wave, cubic for wave breaking). Nightingale's quadratic is closer to the non-wave breaking part (as shown in the figure below). This gives our scheme an even greater advantage in high wind conditions.

(3) However, it is true that using Nightingale's parameterization as non-wave breaking here may cause "double counting" problem.

We added this uncertainty in Model Uncertainty section:

Line 351-357: Nightingale's parameterization was developed from in situ experiments utilizing both volatile and nonvolatile tracers, which potentially incorporate effects from wave breaking. However, they assumed that the transfer velocity only depends on diffusivity without taking solubility effect into consideration. This assumption may not be valid in the presence of breaking waves. Although

our scheme may result in a potential overestimation of the wave-related contributions at lower wind speeds, it has the advantage of distinguishing between different relationships with wind speed—such as linear or quadratic dependencies for non-wave processes and cubic dependencies for wave breaking—particularly at high wind speeds.

[Figure]

Fig. Wind speed dependence of non-breaking transfer velocities. For comparison, other wind speed relationships of the non-breaking transfer velocity calculated by Woolf et al. (2005), Deike and Melville (2018) and Asher et al. (2002) are included (dash line b-d). The k-values are normalized to Schmidt number of 660 (20 °C for $CO_2$ in seawater) and displayed against horizontal wind speed at 10 m [$u_{10}$].

**2. comment:** Line 115 and line 120: I would like the authors to comment on the possible limitation of extrapolating experimental results concerning biological surfactants in the Atlantic Ocean globally. I would like this possible limitation to be mentioned/shortly discussed somewhere in the manuscript (maybe around those very lines, or in the "Model uncertainty" section).
 It's a reasonable suggestion! We discussed the possible limitation in lines 362-368:
Since the composition of surfactant in the Atlantic Ocean may differ from other regions, extrapolating experimental findings on biological surfactants from the Atlantic Ocean to a global scale may introduce uncertainty. There is a risk of underestimating the suppressive effects of surfactants in coastal regions, as shown by Mustaffa et al. (2019), who found that the suppression of kw was in coastal waters compared to oceanic waters. Barthelmess et al. (2022) also showed that refractory DOC from coastal land sources has a more persistent impact on air–sea exchange, while the inhibitory effect of semi-unstable organic matter (dissolved glucose and isoleucine) produced by phytoplankton is stronger but has a shorter impact time.

**3. comment:** Graphical abstract: I salute the authors for having included a nice graphical abstract, which is additional effort but – in my opinion – adds quality to a manuscript. However, I advise the authors to be mindful about making everything well-readable. Currently, most of the text (especially in the lower left corner) is quite hard to read.
We increased the font size of the text and tried to make it more readable.

[Figure]

**4. comment:** Figures 1, 2, and 4: Please increase the font size of the numbers and text in the legend, it is currently very hard to read. It would also be advisable to increase the font size of the panel labelling (a, b, c, d).

We increased the font size of Figures in manuscript and supporting information.

**5. comment:** Lines 16-17: "(…) results in an increase of **over twofold** in the transfer velocity of Hg0 relative to the baseline model." – I feel like there should be a range given, considering that the increase in transfer velocity depends on which of the 4 parametrizations for wave breaking is used.

Thanks for the helpful advice. The sentence in line 18-19 was modified as:

The inclusion of these factors results in an increase of **62-225%** in the global transfer velocity of Hg0 relative to the baseline model.

**6. comment:** Line 57: Please indicate the section of Woolf (1997) where the 30% is mentioned, as I cannot find it explicitly stated. Or was this value derived using figures or tables in Woolf (1997), for instance Table 6.2?

Sorry for the misunderstanding. We derived this value from Table 6.2. They calculate transfer velocity of $CO_2$ with two wind speed distributions. The one calculated by taking Rayleigh distribution has total transfer velocity of $CO_2$ of 21.9 cm h$^{-1}$ and bubble contributed 7.04 cm h$^{-1}$. But the other distribution calculation results in a total sea-air exchange velocity of 41 cm h$^{-1}$ with a bubble contribution of 19.36 cm h$^{-1}$. Here we modified the sentence in line 61 as:

Woolf (1997) estimated that bubbles contribute **about 30-50%** to the global $CO_2$ transfer velocity (…)

**7. comment:** Line 61: It seems like this should be "Reichl", not "Reichel"

Revised as suggested.

**8. comment:** Line 86: The name "Darwin model" was not explicitly mentioned in Dutkiewicz et al., 2012. Was this term later added? Maybe there is some reference missing?

Actually, it may be a project name **(the DARWIN project: http://darwinproject.mit.edu/)**. It's a

global ocean biogeochemistry model developed within the MITgcm framework to simulate the biogeochemical cycle of organic carbon and associated marine plankton ecosystem. We added this website in our manuscript in line 90.

**9. comment:** Line 93: Where does the "doubled" come from? From Loose et al. (2014): "The model indicates that effects from shear and convection in the sea ice zone contribute an additional 40 % to the magnitude of keff, beyond what would be predicted from an estimate of keff based solely upon a wind speed parameterization"

There is a non-linear relationship between sea ice cover and the kw impaction shown in Fig. 10 with approximately 42% on average (Loose et al., 2014). This "double" parameterization of drifting sea ice on Hg was first introduced in MITgcm by Zhang et al. (2015) to get a better simulation of Hg in the Arctic Ocean. And we didn't change it in this manuscript. We revised our Methodology section and added reference in line 102.

**10. comment:** Line 96-97: "(…) is the Henry's law constant (…)" - Should this not be the **dimensionless** Henry's law constant?

Thanks for pointing this out. This is **dimensionless** Henry's law constant. we revised as suggested.

**11. comment:** Line 143: I cannot find an Asher et al. (2002) in the reference list. Please correct.

Sorry for the mistake. The Asher et al. (2002) has been placed in the reference, but there was an error in the year when exporting the references and we changed 2013 to 2002.

**12. comment:** Line 155: A word appears to be missing here.

The sentence in line 162 was modified as: (…) which provides a better **simulation of** the transfer velocity.

**13. comment:** Line 172: Considering that the unit of suppression is in percentage (%) such that 100% = 1, shouldn't this read: (100-suppression [%])/100?

Thanks for pointing out the problem; we have corrected it in the manuscript.

**14. comment:** Table 1: Even if somewhat redundant, it may still be helpful for the reader here to indicate the naming of the parametrizations (e.g AW98, A02) next to the corresponding equations

we added the naming into Table 1. It should be noticed that we simplified the names of the parameterization (SUR1 for surfactants suppression and WB1-4 for wave breaking).

**15. comment:** Line 223: **high** variation

Revised as suggested.

**16. comment:** Line 224-225: Another point where it could be explicitly mentioned that the surfactant parametrization used here was based on data from the Atlantic Ocean, and that extrapolation adds uncertainty.

Thanks for the helpful advice. We added another point in line 216-217:

Our surfactant parametrization used here was based on data from the Atlantic Ocean which may not be applicable to other regions.

Please also refer to our response to comment 2.

**17. comment:** Line 263: Minor detail, but maybe it would be better to write "comparatively small" instead of "relatively small" to make clear that the magnitude of effect on CO2 is evaluated in comparison to the effect on Hg0. I wouldn't call a 30-40% increase in CO2 exchange "relatively small".

Thank you for providing these valuable suggestions regarding my writing. we have corrected it in line 255.

**18. comment:** Line 321: One has to be careful with what is what is ment with "lifetime" here. If it is lifetime against deposition, then it shouldn't be increased by increased atmospheric Hg concentrations. If it is lifetime against permanent burial, then this argument could be made because of more efficient reemission from the ocean. I advise to clearly state here what is ment with "lifetime".

The sentence in line 313 was modified as:

The increased $Hg^0$ evasion may increase atmospheric Hg concentrations and thus Hg deposition, as well as prolong Hg lifetime in the ocean biogeochemical cycles.

**19. comment:** Line 344: I think this should say **REpresent**, not **present**

Revised as suggested.

**20. comment:** Line 346: In **THE** Southern Ocean

Revised as suggested.

**21. comment:** Line 379-380: This sentence feels off at the moment. Maybe "(…) is **CURRENTLY** the only **USED**"?

Appreciate your suggestion. The sentence in line 382-383 was modified as:

The estimation of $Hg^0$ air–sea exchange is of great uncertainty since wind speed is currently the only used.

**22. comment:** Line 387-388: This is weirdly worded at the moment and not clearly understandable, please reformulate.

The sentence in line 389-390 was modified as:

We find that surfactants can reduce the transfer velocity of Hg0 by 0–16.7%. However, wave breaking has a much more significant impact, increasing the transfer velocity by 1–3 times due to the low solubility of Hg.

**23. comment:** Line 391: observation**S**

Revised as suggested.

**24. comment:** Line 392: it "is reduced by concentration change" is unclear. Is something missing here?

As only ocean part is considered, the atmospheric deposition was set as constant. The higher emission will be reduced sea surface $Hg^0$ concentration, and thus reduce air-sea exchange. The sentence in line 394 was modified as:

Hg$^0$ air–sea exchange flux is increased in mid- to high-latitude regions with high wind and wave breaking efficiency (28–56%). Conversely, in low-latitude regions with lower wind speeds and in nearshore areas with reduced wave activity, the flux decreases by 16–31% as the surface concentration of Hg$^0$ diminishes due to higher emissions.

**25. comment:** Line 397: Maybe better: "**even** higher"
Revised as suggested.

**26. comment:** Line 399: I suggest to tone this down a bit, e.g. "may have resulted"
Revised as suggested.

**27. comment:** Line 400: "(…) as the exclusive proxy." - proxy for what? It should be explicitly said. I guess this is the gas exchange velocity.
Thanks for the helpful advice. The sentence in line 404 was modified as:
(…) when using wind speed as the only parameter in the estimation of gas exchange velocity.

**28. comment:** Line 405-408: This is quite a long sentence that might be better split into 2.
The sentence in line 409 was modified as:
Thus, we highlight the necessity for direct high-resolution measurements of Hg$^0$ flux, especially simultaneous observation of other parameter like wave height, surfactant concentration and chemical composition. These data are essential for modelers to develop and validate robust models for simulating the diel, seasonal and inter-annual Hg dynamics on a local to regional scale.

**29. comment:** Supplementary Figure S2: The figure is very useful, but the text is very small. I suggest to just increase the figure size on the page, considering that there is no real space limitation in the supplementary material.
Revised as suggested.

**30. comment:** Supplementary Figure S4: Please increase font size in the figure legend.
Revised as suggested.

**31. comment:** It may be interesting to mention for experimentalists if, for improvement of the parametrizations, observational data from certain oceanic regions would be especially useful, for example because model results in certain regions are especially sensitive to the used parametrization.
According to our modelling, the sea-air exchange flux of Hg has the greatest sensitivity to different wave breaking parameterizations and surfactant parameterization in the North Atlantic (Fig. 4). Though there are currently no simultaneous observations for the various parameters, we believe that any synchronous observations of wave height, surfactants concentration and chemical composition may be helpful to modelers. We added this information in line 411-415:
Because of the high sensitivity to different parameterizations at middle and high latitudes, especially in the North Atlantic (Fig. 4), we believe that synchronous observations in these regions may be helpful for modelers to develop and validate robust models for simulating the diel, seasonal and inter-annual Hg dynamics on a local to regional scale.

**Response to Referee #2:**

This work is the first to evaluate the impact of wave breaking and surfactants on the air-sea exchange of mercury, building on recent advances in the measurement of Hg fluxes and the general understanding of the gas exchange process. The authors inserted new parameterizations into a global coupled model for Hg that had been previously validated, also testing different schemes for solving the wave breaking process. The manuscript is well conceived and presented and the results are of broad interest. Indeed, the flux from the ocean to the atmosphere is considered to be one of the largest in the global Hg cycle but is still not well constrained. The manuscript could be further improved with some adjustments, especially in the methodology section.

We would like to thank the referee for the thoughtful and useful comments. Those comments are all valuable and very helpful for revising and improving our paper, as well as the important guiding significance to our researches. We have studied comments carefully and have made corrections.

**Introduction**

**1. comment:** I suggest adding some background information on the estimated magnitude of global air–sea exchange of Hg and the associated uncertainties to emphasize the knowledge gap and the importance of the research question. A summary of how different model configurations and parameterizations have affected these estimates in previous modeling efforts (e.g., Zhang et al., 2019, 2023) would be interesting.

Thank you very much for the valuable suggestion. We added background information in lines 38:
The estimated magnitude of global air–sea exchange of Hg ranges from to 2840 Mg a$^{-1}$ to 3950 Mg a$^{-1}$ (Zhang et al., 2019; Liss and Merlivat, 1986; Wanninkhof et al., 1992; McGillis et al., 2001; Zhang et al., 2023). Osterwalder et al. (2021) further demonstrated that different transfer velocity parameterizations can lead to more than a fourfold variation in sea-air exchange flux estimates along the coastal Baltic Sea (0.3±0.6 ng m$^{-2}$ h$^{-1}$ to 2.6±0.6 ng m$^{-2}$ h$^{-1}$).

**Methodology**

**2. comment:** Several equations (6, 8, 9, 10, 11, 12 and 13) are found both in the text and in Table 1. I suggest retaining them only in Table 1 and referring to them in the text with equation numbers. Referring to the equations could also reduce some confusion and redundancy elsewhere in the text. For example, in lines 87-103, many repetitions could be avoided by small rearrangements, e.g.:
*The air–sea exchange of Hg0 (eq.1) is calculated from the exchange velocity (kwHg0) and the concentration gradient of Hg0 across the air–sea interface corrected for Henry's law constant (Cw-Ca/H) (Andersson et al., 2008). H [...] quantifies the ability of the dissolved phase to [...].* [This is general, not only for baseline]. *In the **baseline** parameterization, the exchange velocity of Hg0 on the ocean side (eq.2) is estimated following the quadratic relationship with wind speed proposed by Nightingale et al. (2000) **for CO2 (eq. 3)** adjusted for the Schmidt number of Hg0* [add somewhere the eq. of ScHg0 to show the relationship with diffusivity and temperature], *and for the proportion of ice-free sea surface areas.*

That's a valuable suggestion! We revised the Methodology and retained the equations only in Table 1.

**3. comment:** The terms *kwexch, kHg0, kwHg0, and kw* are used somewhat ambiguously in the

manuscript and should be homogenized. It might be convenient to have Eqs. 2 and 14 close together to make it clear that the latter is the modified version of the former. Thus, lines 170-173 might best be inserted after line 103 to introduce subsections 2.2. and 2.3 (where a detailed explanation of the terms introduced in eq. 14 is given). Similarly, lines 175-178 should be placed outside the section on wave breaking. They could be moved together with the previous paragraph or to another subsection dedicated to the simulations (together with lines 104-106).

Still concerning the equations:

- *pisvelo* is a gas transfer velocity normalized to the Schmidt number of CO2 in freshwater at a temperature of 20 °C, called k600 in Nightingale et al. and elsewhere in the literature. I suggest using k600 also in this text.
- in eq. 14 and elsewhere a more concise name could be found for "suppression of kw" (e.g. Ssurf, Skw?)

Thanks for the excellent advice. We homogenized the expression of kw in manuscript by using kw to represent transfer velocity of $CO_2$ and kwHg0 for Hg. We also changed the order of the lines according to your suggestions. The *pisvelo* and "suppression of kw" are revised as suggested.

**4. comment:** As for surfactants (lines 108-110), marine bacteria also seem to be involved in their production (see e.g. Kurata et al. 2016 in Sci Rep), which at least partly explains why TOC is a better proxy than Chla. The marine bacteria are indeed shown in the graphical abstract, but they are neglected in the text. Also, I find it somewhat inaccurate to say that PP is "commonly represented" by Chla. I would rather say that it is "estimated from" Chla for operational reasons (i.e. remote sensing). I suggest to critically revise this paragraph to clarify the current understanding of surfactant dynamics.

It's a reasonable suggestion! We revised the paragraph in lines 113-125:

Surfactants are mainly originated from ocean biological activities (Lin et al., 2002), with elevated concentrations anticipated in regions characterized by increased primary productivity (PP) (Wurl et al., 2011). The concentration of surfactants at the sea surface is related to PP, which is commonly estimated from chlorophyll a (Chl a) (Tsai and Liu, 2003) for operational reasons (i.e. remote sensing). Nevertheless, recent studies have shown that Chl a cannot fully predict the occurrence of surface surfactants when used as a substitute for PP (Wurl et al., 2011; Sabbaghzadeh et al., 2017). Some strains of heterotrophic bacteria are known to produce surfactants (Satpute et al., 2010) and have been linked to a surfactant-covered ocean surface (Kurata et al., 2016). Additionally, the occurrence of surfactants is also subject to influence from meteorological conditions, including solar radiation (Gasparovic et al., 1998) and precipitation (Wurl and Obbard, 2005). Surface tension (Schmidt and Schneide, 2011), organic carbon concentration (Calleja et al., 2009; Barthelmeß et al., 2021), and sea surface temperature (Pereira, 2018) are also used to predict the occurrence of surface surfactants. However, most studies have not provided a clear quantitative relationship.

**Tables**

**5. comment:** The readability of Table 1 must be improved (e.g. add the associated parameterization names for $k_{bub}$, use - or not - horizontal lines consistently in all rows, remove apex letters for references from the equations of $k_{bub}$).

Revised as suggested.

**6. comment:** Table 2. I suggest simplifying the names of the parameterizations (eg. SUR1, WB1, WB2, WB3, WB4?) and distinguishing the sensitivity simulations from the other ones by using different names (e.g., CaseA-CaseC instead of 5-7).

Revised as suggested.

**Results**

**7. comment:** Can the results from Wurl et al., 2011 be used to validate the modeled distribution of surfactants, at least qualitatively?

Thank you very much for the valuable suggestion. They also reported a more significant SML coverage in tropical than north of 30°N and south of 30°S. We added a sentence in lines 201-202: This finding is consistent with that of Wurl (2011) who reported a more significant SML coverage between 30°N and 30°S.

**8. comment:** line 394: the net flux of Hg0 evasion for the baseline simulation is 3841 Mg/a, higher than 3000 Mg/a given in Zhang et al., 2019 for the offline coupled model. Is this due to only the different initial conditions used in the model?

Yes, the atmospheric component of Zhang et al. (2019) used the results of Horowitz et al. (2017). In this paper, Shah's updated scenario is used (Shah et al., 2021), resulting in different initial ocean Hg concentrations and atmospheric deposition.

**Minor Comments**

**9. comment:** line 39 I suggest changing "production" with "activity"

Revised as suggested.

**10. comment:** lines 40-42: I suggest using "they" instead of repeating surfactants and using a comma rather than a period before "Second"

Revised as suggested.

**11. comment:** line 63 reword "more significant for Hg0 with lower solubility."

The sentence in lines 68-69 was modified as: The significance of bubble effects depends on the solubility of gases in seawater. It is anticipated that bubble effects will be more pronounced for Hg0, given its lower solubility.

**12. comment:** line 126 the subsection number needs to be corrected

Revised as suggested.

**13. comment:** line 135 I suggest rewording as follows: […] we attempt to use four different parameterization schemes, all considering the significant wave height (Hs), which has been proved to be [...]

Revised as suggested.

**14. comment:** line 168 add the article a to dimensionless

I am sorry for the mistake here. $A_B$ is an empirical factor with dimensions that fit the observations. We modified the sentence in line 149 as: where $A_B$ is **dimensional** fitting coefficient.

**15. comment:** line 223 highly should be high

Revised as suggested.

**16. comment:** line 224 therefore, the suppression relationship may not be linear?

Thank you for pointing this out. Indeed, the suppression relationship may not be linear. We modified the sentence in line 215 as: Therefore, the suppression relationship may change in different

environments.

**17. comment:** line 280 please specify stronger than …

Osterwalder et al. (2021) reported the cubic gas transfer velocity of $Hg^0$ by using eddy covariance flux measurements. A higher index value indicates that $Hg^0$ is more sensitive to changes in wind speed than $CO_2$. We modified the sentence in line 271 as: Recent research has shown that the transfer velocities of $Hg^0$ **are more sensitive to** wind speed (with higher index) by using eddy covariance flux measurements.

**18. comment:** line 294 but lower than

Revised as suggested.

**19. comment:** line 362 another verb would fit best than implied

We modified the sentence in line 367 as: Barthelmess et al. (2022) also showed that […]

**20. comment:** line 365 revise "The highly spatial-temporal variations in short-term and seasonal of surfactants"

**21. comment:** line 366 chemical composition has been already discussed at lines 359-361

We merge the sentences in lines 359-361 with line 366 as: The parameterization of the surfactant suppression is quite challenging, because significant spatial-temporal variations in surfactants and changes in the chemical composition of surfactants may affect the relationship between TOC concentration and surfactant concentration.

**22. comment:** line 380 the only parameter considered.

The sentence in line 382 was modified as: The estimation of $Hg^0$ air−sea exchange is of great uncertainty since wind speed is currently the only used.

**23. comment:** line 384 sensitivity

Revised as suggested.

**24. comment:** line 398 the word research is uncountable

Revised as suggested.

**25. comment:** line 391 add an s to observation

Revised as suggested.

**Reference**

Asher, W., Edson, J., Mcgillis, W., Wanninkhof, R., Ho, D. T., and Litchendor, T.: Fractional Area Whitecap Coverage and Air-Sea Gas Transfer Velocities Measured During GasEx-98, in: Geophysical Monograph Series, edited by: Donelan, M. A., Drennan, W. M., Saltzman, E. S., and Wanninkhof, R., American Geophysical Union, Washington, D. C., 199–203, https://doi.org/10.1029/GM127p0199, 2002.

Barthelmeß, T. and Engel, A.: How biogenic polymers control surfactant dynamics in the surface microlayer: insights from a coastal Baltic Sea study, Biogeosciences, 19, 4965–4992, https://doi.org/10.5194/bg-19-4965-2022, 2022.

Deike, L. and Melville, W. K.: Gas Transfer by Breaking Waves, Geophysical Research Letters, 45, https://doi.org/10.1029/2018GL078758, 2018.

Fairall, C. W., Yang, M., Bariteau, L., Edson, J. B., Helmig, D., McGillis, W., Pezoa, S., Hare, J. E., Huebert, B., and Blomquist, B.: Implementation of the Coupled Ocean-Atmosphere Response Experiment flux algorithm with $CO_2$, dimethyl sulfide, and $O_3$, J. Geophys. Res., 116, C00F09, https://doi.org/10.1029/2010JC006884, 2011.

Gašparović, B., Kozarac, Z., Saliot, A., Ćosović, B., and Möbius, D.: Physicochemical

Characterization of Natural and *ex-Situ* Reconstructed Sea-Surface Microlayers, Journal of Colloid and Interface Science, 208, 191–202, https://doi.org/10.1006/jcis.1998.5792, 1998.

Horowitz, H. M., Jacob, D. J., Zhang, Y., Dibble, T. S., Slemr, F., Amos, H. M., Schmidt, J. A., Corbitt, E. S., Marais, E. A., and Sunderland, E. M.: A new mechanism for atmospheric mercury redox chemistry: implications for the global mercury budget, Atmos. Chem. Phys., 17, 6353–6371, https://doi.org/10.5194/acp-17-6353-2017, 2017.

Kurata, N., Vella, K., Hamilton, B., Shivji, M., Soloviev, A., Matt, S., Tartar, A., and Perrie, W.: Surfactant-associated bacteria in the near-surface layer of the ocean, Sci Rep, 6, https://doi.org/10.1038/srep19123, 2016.

Lin, I. -I., Wen, L., Liu, K., Tsai, W., and Liu, A. K.: Evidence and quantification of the correlation between radar backscatter and ocean colour supported by simultaneously acquired in situ sea truth, Geophysical Research Letters, 29, https://doi.org/10.1029/2001gl014039, 2002.

Liss, P. S. and Merlivat, L.: Air-Sea Gas Exchange Rates: Introduction and Synthesis, in: The Role of Air-Sea Exchange in Geochemical Cycling, edited by: Buat-Ménard, P., Springer Netherlands, Dordrecht, 113–127, https://doi.org/10.1007/978-94-009-4738-2_5, 1986.

McGillis, W. R., Edson, J. B., Ware, J. D., Dacey, J. W. H., Hare, J. E., Fairall, C. W., and Wanninkhof, R.: Carbon dioxide flux techniques performed during GasEx-98, Marine Chemistry, 75, 267–280, https://doi.org/10.1016/S0304-4203(01)00042-1, 2001.

Mustaffa, N. I. H., Ribas-Ribas, M., Banko-Kubis, H. M., and Wurl, O.: Global reduction of *in situ* $CO_2$ transfer velocity by natural surfactants in the sea-surface microlayer, Proc. R. Soc. A., 476, 20190763, https://doi.org/10.1098/rspa.2019.0763, 2020.

Osterwalder, S., Nerentorp, M., Zhu, W., Jiskra, M., Nilsson, E., Nilsson, M. B., Rutgersson, A., Soerensen, A. L., Sommar, J., Wallin, M. B., Wängberg, I., and Bishop, K.: Critical Observations of Gaseous Elemental Mercury Air-Sea Exchange, Global Biogeochemical Cycles, 35, https://doi.org/10.1029/2020GB006742, 2021.

Sabbaghzadeh, B., Upstill-Goddard, R. C., Beale, R., Pereira, R., and Nightingale, P. D.: The Atlantic Ocean surface microlayer from 50°N to 50°S is ubiquitously enriched in surfactants at wind speeds up to 13 m s $^{-1}$: Atlantic Ocean Surfactants, Geophys. Res. Lett., 44, 2852–2858, https://doi.org/10.1002/2017GL072988, 2017.

Shah, V., Jacob, D. J., Thackray, C. P., Wang, X., Sunderland, E. M., Dibble, T. S., Saiz-Lopez, A., Černušák, I., Kellö, V., Castro, P. J., Wu, R., and Wang, C.: Improved Mechanistic Model of the Atmospheric Redox Chemistry of Mercury, Environ. Sci. Technol., 55, 14445–14456, https://doi.org/10.1021/acs.est.1c03160, 2021.

Wanninkhof, R.: Relationship between wind speed and gas exchange over the ocean, J. Geophys. Res., 97, 7373, https://doi.org/10.1029/92JC00188, 1992.

Woolf, D. K.: Parametrization of gas transfer velocities and sea-state-dependent wave breaking, Tellus B: Chemical and Physical Meteorology, 57, 87, https://doi.org/10.3402/tellusb.v57i2.16783, 2005.

Wurl, O. and Obbard, J. P.: Chlorinated pesticides and PCBs in the sea-surface microlayer and seawater samples of Singapore, Marine Pollution Bulletin, 50, 1233–1243, https://doi.org/10.1016/j.marpolbul.2005.04.022, 2005.

Wurl, O., Wurl, E., Miller, L., Johnson, K., and Vagle, S.: Formation and global distribution of sea-surface microlayers, Biogeosciences, 8, 121–135, https://doi.org/10.5194/bg-8-121-2011, 2011.

Zhang, Y., Jacob, D. J., Dutkiewicz, S., Amos, H. M., Long, M. S., and Sunderland, E. M.:

Biogeochemical drivers of the fate of riverine mercury discharged to the global and Arctic oceans, Global Biogeochemical Cycles, 29, 854–864, https://doi.org/10.1002/2015GB005124, 2015.

Zhang, Y., Horowitz, H., Wang, J., Xie, Z., Kuss, J., and Soerensen, A. L.: A Coupled Global Atmosphere-Ocean Model for Air-Sea Exchange of Mercury: Insights into Wet Deposition and Atmospheric Redox Chemistry, Environ. Sci. Technol., 53, 5052–5061, https://doi.org/10.1021/acs.est.8b06205, 2019.

Zhang, Y., Zhang, P., Song, Z., Huang, S., Yuan, T., Wu, P., Shah, V., Liu, M., Chen, L., Wang, X., Zhou, J., and Agnan, Y.: An updated global mercury budget from a coupled atmosphere-land-ocean model: 40% more re-emissions buffer the effect of primary emission reductions, One Earth, 6, 316–325, https://doi.org/10.1016/j.oneear.2023.02.004, 2023.